# SARS-CoV-2 Infection in Patients with Cystic Fibrosis: What We Know So Far

**DOI:** 10.3390/life12122087

**Published:** 2022-12-13

**Authors:** Carmelo Biondo, Angelina Midiri, Elisabetta Gerace, Sebastiana Zummo, Giuseppe Mancuso

**Affiliations:** 1Department of Human Pathology, University of Messina, 98125 Messina, Italy; 2ASP (Azienda Sanitaria Provinciale), 90141 Palermo, Italy

**Keywords:** cystic fibrosis, viral infections, COVID-19

## Abstract

Respiratory infections are the most common and most frequent diseases, especially in children and the elderly, characterized by a clear seasonality and with an incidence that usually tends to decrease with increasing age. These infections often resolve spontaneously, usually without the need for antibiotic treatment and/or with the possible use of symptomatic treatments aimed at reducing overproduction of mucus and decreasing coughing. However, when these infections occur in patients with weakened immune systems and/or underlying health conditions, their impact can become dramatic and in some cases life threatening. The rapid worldwide spread of Severe Acute Respiratory Syndrome Coronavirus 2 (SARS-CoV-2) infection has caused concern for everyone, becoming especially important for individuals with underlying lung diseases, such as CF patients, who have always paid close attention to implementing protective strategies to avoid infection. However, adult and pediatric CF patients contract coronavirus infection like everyone else. In addition, although numerous studies were published during the first wave of the pandemic on the risk for patients with cystic fibrosis (CF) to develop severe manifestations when infected with SARS-CoV-2, to date, a high risk has been found only for patients with poorer lung function and post-transplant status. In terms of preventive measures, vaccination remains key. The best protection for these patients is to strengthen preventive measures, such as social distancing and the use of masks. In this review, we aim to summarize and discuss recent advances in understanding the susceptibility of CF individuals to SARS-CoV-2 infection.

## 1. Introduction

An outbreak of pneumonia associated with a novel coronavirus (2019-nCoV) was first identified in Wuhan, Hubei Province, China, on 8 December 2019 [1]. The coronavirus most likely jumped from a wild animal in a cage to people attending a wholesale seafood market [2]. This new coronavirus responsible for the outbreak was identified by deep sequencing analysis of the virus genome isolated from lower respiratory tract samples [3]. While initially only people who had frequented the wholesale seafood market tested positive for severe acute respiratory syndrome coronavirus 2 (SARS-CoV-2) infection between November 2019 and December 2020, later (late January and February 2020), this positivity for infection was extended to people who had never been to the seafood market, suggesting possible person-to-person transmission of this novel coronavirus [4,5,6,7]. As the epidemic progressed, infections were reported in other provinces in China in the second week of January 2020 and subsequently in other countries including Singapore, the Republic of Korea, and Japan [8]. The World Health Organization (WHO) declared COVID-19 a global health emergency on 31 January 2020, and, in view of the severity of the epidemic and its ability to spread rapidly internationally, declared COVID-19 a pandemic on 12 March 2020 [9]. According to WHO as of 27 September 2020, COVID-19 had sickened more than 32 million people and caused nearly one million deaths worldwide [10]. Some countries have experienced a resurgence of COVID-19 [11,12]. Although initially, the epidemic seemed under control after timely public health interventions, as the number of infections increased, several waves of infection occurred, triggering a series of blocking measures [13,14]. Researchers promptly attempted to map the epidemiology of cases during the early phase of the pandemic but the presence of multiple genetic clusters of SARS-CoV-2 led to important changes in the epidemiology of COVID-19 [14,15]. The disease course of COVID-19 is governed by important epidemiological parameters whose understanding is critical to guide interventions. Among the most important parameters are containment delay (time between symptom onset and mandatory isolation) and serial intervals (time between symptom onset of infector-infect pairs) [16,17]. Data published by WHO show that by the end of October 2022, there had been more than 626 million confirmed cases of COVID-19, including more than 6.5 million deaths, with a total of nearly 13 billion doses of vaccines administered [18].

### Cystic Fibrosis Lung Disease

Cystic fibrosis (CF) is a chronic genetic disease that causes severe damage to the lungs and other organs in the body [19,20]. Although chronic pulmonary infection with *Pseudomonas aeruginosa* is the leading cause of morbidity and mortality in CF patients, viral respiratory infections also pose a serious danger because of the risk of acute exacerbations with increased short- and long-term morbidity [21,22]. Since SARS-CoV-2 caused a pandemic respiratory disease called COVID-19, especially in individuals with comorbidities, people with CF are considered a clinically “fragile” population with a higher inherent risk of chronic lung function deterioration and development of severe COVID-19 than healthy people [23,24,25,26]. Consequently, since the beginning of the pandemic, people with CF have been advised to increase their level of protection with screens and self-isolation to avoid the risk of cross-infection [27,28,29].

Moreover, a remote patient-monitoring program has been implemented to check their clinical conditions and provide a constantly updated treatment plan [30,31]. Surprisingly, contrary to all expectations, the incidence of SARS-CoV-2 in the CF population was lower than the average incidence in the general population [32,33]. From the data collected, although not conclusive, it seems clear that few individuals with CF tested positive for SARS-CoV-2 infection (probably because they were asymptomatic or had mild disease) and that, among those infected, only very few had severe outcomes [10,23,34]. Therefore, the primary objective of this review was to provide a summary of the available scientific evidence on the susceptibility of people with CF to SARS-CoV-2 infection. The secondary objective was to compare the prevalence of COVID-19 and disease severity in CF patients with those in the general population.

## 2. Methods

A literature review was undertaken from August 2020 until September 2022, using the following search terms: “cystic fibrosis,” “SARS-CoV-2,” “COVID-19,” on the PubMed databases. Eight studies were included in this review including three retrospective studies. Only articles written in English and published in peer-reviewed journals were reviewed.

### 2.1. The Complexity of SARS-CoV-2 Infection

#### 2.1.1. SARS-CoV-2 Origin and Classification

The 2019 novel CoV (SARS-CoV-2), initially named 2019 novel coronavirus (2019-nCoV), belongs to human CoVs (HCoVs) [35]. To date, in addition to severe acute respiratory syndrome coronavirus 2 (SARS-CoV-2), six human coronaviruses (HCoVs) have been identified: 229E, OC43, HKU1, NL63, SARS-CoV, and Middle East respiratory syndrome (MERS) [36]. All HCoVs are enveloped non-segmented, positive-sense RNA viruses that can affect different species, causing a wide spectrum of diseases [37,38]. All of these viruses have a zoonotic origin (bats, rodents, or domestic animals) and according to serology studies and genomic analysis, HCoV-229E and HCoV-NL63 belong to the genus *Alphacoronavirus* while HCoV-OC43 and HCoV-HKU1 are members of *Betacoronavirus* [1,39]. All four of these HCoVs (229E, NL63, OC43, and HKU1) are widespread globally and mostly cause mild respiratory illness (contributing to 15–30% of cases of common colds in humans), as they mainly infect the upper respiratory tract, especially during the winter months [40,41]. In contrast to these four common HCoVs, the other three coronaviruses (SARS-CoV, MERS-CoV, and SARS-CoV-2) are associated with severe respiratory disease because they can spread to the lower respiratory tract and cause severe pneumonia with mortality rates of about 10% and 30% for the SARS-CoV and MERS-CoV, respectively [42,43].

The first human coronavirus (HCoV) was isolated in 1965 [44]. Since then, more than 30 additional strains were identified, including the SARS-CoV-2 virus that was reported in Wuhan, the capital of Hubei, China, in late 2019. This pathogen then rapidly spread to all over China and the world, causing severe pneumonia that, in a subset of patients, led to rapidly worsening respiratory failure and acute respiratory distress syndrome (ARDS), often requiring endotracheal intubation and mechanical ventilation [45,46,47,48]. COVID-19 caused more than five million deaths in two years, although the global death rate, due to the high transmissibility and pathogenicity of this virus, could be double or even quadruple [49,50]. In addition, several studies have shown that health workers were at greater risk of mental health problems due to the enormous pressure experienced during the COVID-19 pandemic.

#### 2.1.2. SARS-CoV-2 Morphology and Replication

Like other coronaviruses, SARS-CoV-2 is an enveloped virus with a non-segmented, single-stranded, positive-sense RNA filament [51,52]. The virus contains a nucleocapsid with helical symmetry and its viral particles are composed of four structural proteins: glycoprotein S (spike), transmembrane protein M (membrane), envelope protein E (envelope), and nucleoprotein N (nucleocapsid) [51]. In addition to these structural proteins, the RNA genome of SARS-CoV-2 encodes 15 non-structural proteins and eight accessory proteins that play important roles in virus replication and assembly processes [53,54]. Cell entry of SARS-CoV-2 is mediated by the binding of the S protein of the virus to the host angiotensin-2–converting enzyme (ACE-2) [55]. This cellular receptor is a type I membrane glycoprotein responsible for the conversion of angiotensin II into angiotensin 1–7 present on the plasma membrane of airway epithelial cells, goblet secretory cells, and type II pneumocytes [56,57,58]. Protein S binding to ACE-2 triggers a cascade of events leading to the fusion of viral and cell membranes, resulting in capsid release into the cytosol [59] (Figure 1). SARS-CoV-2 replication is a complex process that occurs entirely in the cytoplasm and involves the action of several viral and host proteins to perform RNA polymerization, proofreading, and capping [60,61].

#### 2.1.3. SARS-CoV-2 Infection

The first macromolecular event after virus entry is the release and uncoating of incoming genomic RNA, which is immediately translated into two polyproteins that post-translationally process into the non-structural proteins that form the viral replication and transcription complex [59,62,63]. Simultaneously with the expression of these proteins, SARS-CoV-2 reorganizes the membranes of the endoplasmic reticulum to form a reticulovesicular network consisting of the characteristic perinuclear double-membrane vesicles (DMVs), convoluted membranes (CMs), and small open double-membrane spherules (DMSs), which create a protective microenvironment for genomic RNA replication and the generation of a nested array of subgenomic mRNAs [63,64,65]. Assembly of mature SARS-CoV-2 virions occurs within the ER-to-Golgi intermediate compartment while virions are secreted from the infected cell by exocytosis [10,66] (Figure 1). After release of the viral genome into the cytoplasm of the cell, several signaling cascades are activated, inducing an intense pro-inflammatory response [10,67]. Viral infection in airway epithelial cells can cause high levels of pyroptosis, a form of programmed cell death, that results in increased secretion of pro-inflammatory cytokines and chemokines such as IL-6, IFNγ, MCP1, and IP-10 [68,69,70,71].

### 2.2. Viral Respiratory Infections in Cystic Fibrosis

CF is an inherited, lethal, recessive genetic disease affecting about 200,000 people in the U.S., most commonly in Caucasians, caused by mutations in the Cystic Fibrosis Transmembrane Regulator (CFTR) gene [72]. Since the discovery of the CFTR gene in 1989, more than 2000 CFTR gene variants have been reported, of which more than 350 have confirmed correlation with disease and another 46 have shown variable clinical effects [73,74]. The most common mutation in the CFTR gene that causes cystic fibrosis involves the deletion of three base pairs encoding a phenylalanine residue at position 508 on chromosome 7 (Phe508; Δ508) [75]. People with CF have reduced survival due to progressive loss of lung function secondary to chronic airway infection [76]. The estimated median age of survival from birth was 46 years in males and 41 years in females for homozygous Phe508del patients [19,77]. CF is a multiorgan disease that also impacts the endocrine compartment of the pancreas and leads to exocrine pancreatic insufficiency, resulting in malabsorption, malnutrition, and impaired growth [78,79]. Other clinical features include excessive salt loss in sweat, liver and pancreatic dysfunction, male infertility, sinusitis, and diabetes [20,80]. Mutations that impair CFTR function affect ion transport across cell membranes of airway epithelial cells, leading to a marked reduction in chloride and bicarbonate secretion, which in turn results in a reduction in airway surface fluid volume and pH with an increase in mucus viscosity [81,82,83]. Chronic mucus hyper secretion leads to decreased mucociliary clearance and is a potential risk factor for accelerated loss of lung function [20,82,84]. The intense pulmonary inflammation that characterizes cystic fibrosis persists and involves several immune pathways involved in cytokines production (IL-8 and TNF-α), as well as polymorphonuclear leukocytes (PMNs) accumulation and associated release of proteases such as elastase that cleave the extracellular matrix [20,85,86,87]. In addition, mutations in CFTR also negatively affect the innate immune function of the epithelium by activating the NF-κB pathway by an increased IL-8 production and neutrophil accumulation [88,89]. In addition, defective CFTR function in airway epithelial cells results in reduced surface expression of toll-like receptor-4 (TLR4), leading to downregulation of type I IFN and impaired activation of pulmonary dendritic cells (DCs), which are critical for immune surveillance and coordinated signaling by T cells [86,90,91,92,93]. Therefore, CF patients are susceptible to chronic lung inflammation and infection most commonly due to *Pseudomonas aeruginosa* and *Staphylococcus aureus* [94]. The adaptation of these two pathogens to the CF environment facilitates colonization of this habitat by other infectious agents, resulting in a rapid decline of lung function and reduced survival [95,96]. Among the latter, an important role is played by respiratory viruses, which are a common cause of pulmonary exacerbations in CF patients [21,28]. Previous studies have shown that influenza A and B and rhinoviruses are commonly detected during pulmonary exacerbations in CF [24,97]. During the 2009 flu season, the H1N1 influenza pandemic caused a significant morbidity in most CF patients (e.g., more than 50% required intravenous antibiotic therapy and about 50% were hospitalized) [98,99]. In addition, six patients were admitted to an intensive care unit and three died during hospitalisation [99]. Respiratory syncytial virus (RSV) is also an important cause of acute upper respiratory morbidity in infants with CF [100]. Recent evidence has suggested that bacterial infection may induce changes in the host mucosa and modulate several immune responses, including that related to the reduction of the interferon response to rhinovirus, from which an increase in viral load results [25,101]. In contrast, evidence for the existence of virus–host mucosal interactions remains scarce, and therefore viral infections are not believed to play a significant role in the exacerbation and progression of lung disease in CF patients [21,22].

#### SARS-CoV-2 Infection in Cystic Fibrosis

Because people with CF have chronic deterioration of lung function and are more susceptible to viral infections, they are assumed to beat a greater risk of worse outcomes in case of SARS-CoV-2 infection than healthy individuals [32,102]. However, the real impact of SARS-CoV-2 on CF lung disease is unknown and people with CF may not have a higher risk of acquiring SARS-CoV-2 infection or have worse clinical outcomes in case of infection [23,31,103,104]. A multinational report in eight countries (Australia, The Netherlands, Canada, Ireland, France, New Zealand, USA, and UK) reported data on 40 CF patients with SARS-CoV2 infection [33]. The median age of the patient cohort was 33 years and 43% of this cohort was male. Most (31 out 40) (78%) had generally mild respiratory symptoms, 13 (33%) required oxygen therapy, 4 (10%) were admitted to an intensive care unit, and only one patient (0.4%) required invasive ventilation [33]. These data are in line with the response of the general population. Similarly, a separate retrospective observational COVID-19-CF survey of 20 patients from four European countries showed that the incidence of SARS-CoV-2 in the CF population (0.07%) would appear to be lower than the average incidence in the general population (0.15%) [104,105]. Moreover, a lower incidence was also observed in two other retrospective studies: one from Veneto (Italy) that reported infection rates of 1/532 (0.19%) and 19,729/4,907,704 (0.4%) for people with CF and general population, respectively, [23]; the other was from Spain that reported a cumulative incidence of positive tests for SARS-CoV-2 of 0.49% (49/10,000) in the general population and 0.32% (32/10,000) in CF patients, respectively [106]. More recently, the Global Registry Harmonization Group (GRHG) reported a multinational study conducted by recruiting 181 CF patients from 19 countries who were diagnosed with SARS-CoV-2 [27]. The results of this study clearly show that the outcome of SARS-CoV-2 infection in people with CF has a similar spectrum of outcomes as that observed in the general population. However, a more severe clinical course was observed in CF patients with previous lung function impairment and/or post-transplant status [26,27].

Another prospective study examined the seroprevalence of SARS-CoV-2 in a Belgian cohort of 149 CF patients assessing SARS-CoV-2 antibodies (IgG/IgM) on blood samples [107]. Since the seroprevalence of SARS-CoV-2 among CF patients was still lower than that of the Belgian population, 2.7 percent and 4.3 percent, respectively, the authors of this study believe that this result probably reflects the low incidence of COVID-19 in this population [107].

Similarly, recent studies suggest no difference in the frequency of SARS-CoV-2 infection between children with CF and healthy controls [108,109,110]. Specifically, two different studies were conducted by the GRHG and the European Respiratory Society (ERS) and involved the collection of data from 105 children with CF from 13 countries and 174 ERS centers, respectively [110]. Both authors of these studies concluded that children with CF have a COVID-19 disease course similar to that of the general population [26,111]. Several hypotheses have been formulated to explain these better-than-expected outcomes related to SARS CoV2 infection in CF patients, such as: protective effects of CF therapies (e.g., CFTR modulators, corticosteroids, or azithromycin), a relatively young median age of CF populations, lower prevalence of obesity, adoption of shielding, and protective self-isolation [112,113,114].

Corticosteroids are not directly antiviral, but they act to reduce pro-inflammatory genes, thereby attenuating the immune and inflammatory response [115]. They are often prescribed as adjuvant treatment for CF patients and have been shown to be effective in cases of severe COVID-19 [34]. However, according to other studies, corticosteroids did not improve outcomes in patients with a milder form of COVID-19 [116,117].

Azithromycin is a second-generation macrolide antibiotic used to treat various bacterial infections [118]. In addition, this macrolide has anti-inflammatory and antiviral properties against several respiratory viruses, such as syncytial and influenza viruses [119,120]. Because the use of azithromycin has been associated with cardiac complications and its use for the treatment of COVID-19 is controversial, consequently, this macrolide is not recommended for the treatment of COVID-19 in the absence of an underlying bacterial infection [119,121]. Recent evidence indicates that CFTR modulators may benefit patients with COVID-19 [30,122]. These modulators act by increasing airway fluid production and reducing the pro-inflammatory cytokines responses [122]. Since the inflammatory response of COVID-19 leads to CF-like physiological changes, a positive effect of these modulators on COVID-19 patients is equally possible [123].

Recent studies have reported a potential role of CFTR in the regulation of SARS-CoV-2 infection/replication, which could account for the reduced prevalence of SARS-CoV-2 in people with CF [124,125]. Specifically, these studies have shown that (i) SARS-CoV-2 replication and infection are significantly reduced in CFTR-deleted bronchial cells compared with wild-type cell lines; (ii) ACE-2 expression in CFTR-modified human bronchial epithelia is downregulated compared with that in wild-type cells; and (iii) inhibition of CFTR channel function correlates with the inhibition of viral replication [124,126]. Although several mechanisms may be involved in the inhibition of viral replication in CFTR-defective cells, from the available data, ACE-2 regulation and ion dysregulation appear to be two key mechanisms (Lotti). These data are in agreement with previous data that reported that the severity, number of cases of infection, and viral spread of SARS-CoV-2 in people with CF are significantly lower than those in the normal population [26,105].

Furthermore, people with CF might have a lower incidence of COVID-19 than the general population because, from the beginning of the pandemic, they have been encouraged to reinforce the preventive measures already established in this population, such as walking away, wearing masks, and doing homework to avoid the risk of cross-infection (Figure 2) [29,33].

The figure summarizes the impact of the SARS-CoV-2 pandemic on cystic fibrosis. Several factors have been reported as protective against SARS-CoV2 in CF patients, such as: the use of long-term therapies, “shielding,” “protective self-isolation,” and the lower average age of the CF population.

In addition, to prevent the spread of the virus among CF patients, cystic fibrosis centers not only strengthened recommendations on preventive measures (e.g., avoiding group activities such as gym or dance), but also canceled routine CF clinics and only conducted urgent visits (e.g., respiratory function tests were discontinued) [31,127,128]. Remote assistance (i.e., phone calls and e-mail contacts) was used to monitor patients’ clinical conditions and provide a treatment plan to track progress and make treatment changes when necessary [128]. The multidisciplinary care that characterizes the CF clinic has been successfully converted to a telemedicine model care and clinical appointments have evolved by scheduling patients for in-person or virtual visits [129]. Moreover, clinics have implemented home health monitoring devices and accelerated the use of telemedicine in typical CF care applications (e.g., spirometry, respiratory cultures, diabetes management, mental health care) [130,131]. Although remote care is now an important reality of CF care, it cannot alone support the management of some critical aspects of managed care from the pre-pandemic model of care that contributed to significant gains in health life expectancy [130,132].

## 3. Conclusions

Cystic fibrosis is the most common lifelong recessive genetic disease that, due to the accumulation of mucus that obstructs the airways and traps germs, causes severe damage in several organs, including the lungs, gastrointestinal tract, and pancreatic ducts. Although progressive damage to the respiratory system, typical of CF, is among the conditions that make these patients more at risk for COVID-19, the cumulative incidence of COVID-19 in people with CF is similar to that in the general population. Overall, CF patients with poorer lung function and post-transplant status are at higher risk for worse clinical outcomes in coronavirus disease. Several reasonable hypotheses have been proposed to explain this phenomenon, such as specific drug therapies, azithromycin (anti-inflammatory effect) and DNase (mucolytic effect), characteristics of the CF cohort (younger age, shielding, and protective self-isolation), alteration of intracellular ions that negatively affects virus replication, and lower levels of IL-6 and the cellular protease TMPRSS2 that SARS-CoV-2 uses to enter target cells. However, it is likely that the lower number of infected CF patients is related to increased awareness of infection prevention and control practices. In addition, due to social distancing measures, CF patients were encouraged to work from home and to avoid group activities. In addition, telemedicine was encouraged and routine visits were cancelled to avoid viral spread among CF patients. The COVID-19 pandemic is far from over, and vaccination remains the most important tool for CF patients, with particular priority on developing appropriate vaccination strategies for lung transplant recipients. As the COVID-19 pandemic is a rapidly evolving situation and there are currently no firm data on specific therapies and immunizations for COVID-19 in CF patients, further studies are needed to highlight the major risk factors for severe COVID-19 in CF patients.

## Figures and Tables

**Figure 1 life-12-02087-f001:**
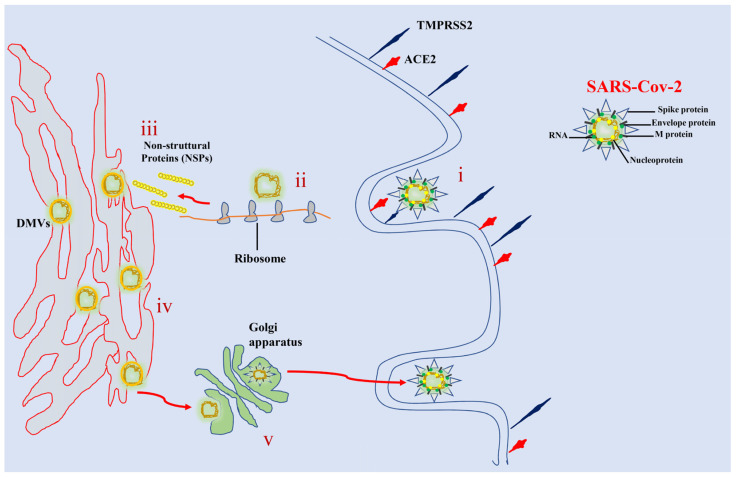
A schematic representation of the SARS-CoV-2 replication cycle. (i) Specific binding between viral S-glycoproteins and the cellular ACE2 receptor mediates entry of SARS-CoV-2 into cells. Spike glycoprotein is cleaved by TMPRSS2 facilitating fusion between the host cell membrane and the virus envelope. (ii) Viral RNA is translated into nonstructural proteins (NSPs). (iii) The host cell machinery is used by the virus to translate viral proteins. (iv) The RNA genome is replicated and translated into bubble-like endoplasmic structures called double membrane vesicles (DMVs). (v) The mature virion is transferred to the Golgi apparatus, where it is released by exocytosis.

**Figure 2 life-12-02087-f002:**
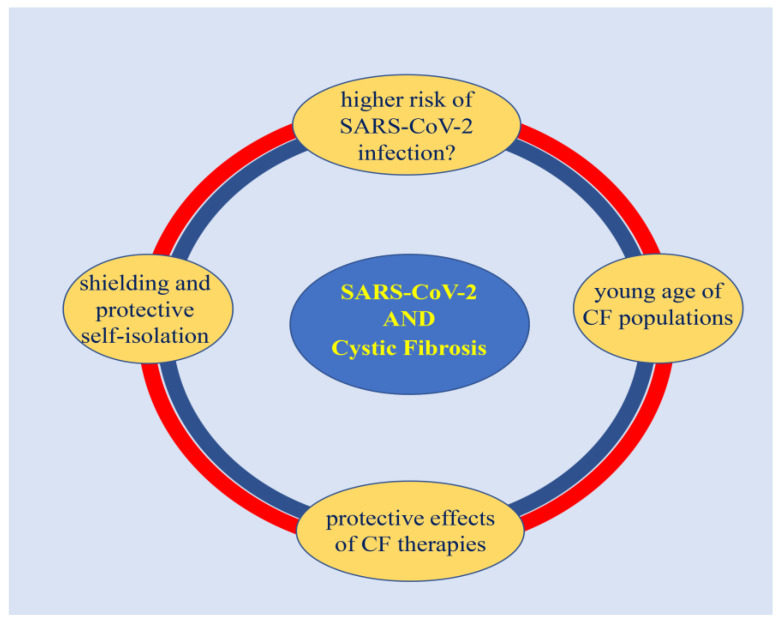
Impact of COVID-19 pandemic on cystic fibrosis.

## Data Availability

Not applicable.

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
