# Peer review of "SARS-CoV-2 Infection in Patients with Cystic Fibrosis: What We Know So Far"

_life, 2022, doi:10.3390/life12122087_

Round 1

Reviewer 1 Report

In the present manuscript, the authors reviewed the current knowledge on the susceptibility of people with cystic fibrosis to SARS-CoV-2 infection and the comparison of prevalence and severity of COVID -19 disease with the general population. Overall, the manuscript is well structured and presented and addresses an interesting topic, namely how the SARS-CoV-2 virus affects people with impaired respiratory function, such as people with cystic fibrosis.

Here are my comments that might help to further improve the manuscript.  

Since the onset of the disease COVID -19, several strains of SARS-CoV-2 have emerged and produced different outcomes, with certain types being more contagious, others causing more severe courses, etc. Have the authors found any information on how the different strains affect people with CF?

Line 163-165: “This cellular receptor is a type I membrane glycoprotein responsible for the conversion of angiotensin II into angiotensin 1–7 present on the plasma membrane of airway epithelial cells, goblet secretory cells and type II pneumocytes [57,58].” What does 1-7 stand for?

Figure 1: The resolution of Figure 1 needs to be improved because it is poorly visible. If the structural parts of the virus are marked, please mark them all. Also, the caption in point ii states that the fusion between the host and virus membranes is shown. However, when you look at the figure, you see the RNA molecule inside the host cell. Please correct this so that it is consistent.

Line: 242-243: “These studies also suggested that viral infections might prevent Pseudomonas colonization in these patients [102,103].” How do they prevent bacterial colonization?

In the section SARS-CoV-2 infection in cystic fibrosis, the authors discuss reports of CF patients with SARS-CoV-2 from 8 countries. Later, they discuss patients from 4 European countries. Are these countries part of the original 8 countries? Or are they different data. Please correct the text to clarify whether these are common or separate data.

The abbreviation CFTR needs to be explained. It should also be explained what the mechanism of CFTR is in SARS-CoV-2 infection/replication. The authors reported that the absence and changes of CFTR affect SARS-CoV-2 replication/infection, but they did not describe the mechanism for this

Figure 2 missing.

Author Response

We would like to thank the reviewers since their comments have greatly helped us to improve the manuscript which was carefully revised according to their suggestions.

Reviewer n.1

Since the onset of the disease COVID -19, several strains of SARS-CoV-2 have emerged and produced different outcomes, with certain types being more contagious, others causing more severe courses, etc. Have the authors found any information on how the different strains affect people with CF?

We agree with the Reviewer that it is important to determine the effect of different strains of the virus on people with CF, but at present these data are not yet available.

Line 163-165: “This cellular receptor is a type I membrane glycoprotein responsible for the conversion of angiotensin II into angiotensin 1–7 present on the plasma membrane of airway epithelial cells, goblet secretory cells and type II pneumocytes [57,58].” What does 1-7 stand for?

Angiotensin-(1–7) is a heptapeptide angiotensin-derived with anti-inflammatory properties. A new reference has been added to the revised manuscript that better explains the role of the Angiotensin-(1–7).

Figure 1: The resolution of Figure 1 needs to be improved because it is poorly visible. If the structural parts of the virus are marked, please mark them all. Also, the caption in point ii states that the fusion between the host and virus membranes is shown. However, when you look at the figure, you see the RNA molecule inside the host cell. Please correct this so that it is consistent.

Thank you to noting this. The resolution of Figure 1 and all other points indicated have been changed.

Line: 242-243: “These studies also suggested that viral infections might prevent Pseudomonas colonization in these patients [102,103].” How do they prevent bacterial colonization?

Thank you to noting this. This statement is clearly incorrect because Pseudomonas is known to block the antiviral response of airway epithelial cells, promoting the spread of the virus. The sentence has been eliminated in the revised manuscript.

In the section SARS-CoV-2 infection in cystic fibrosis, the authors discuss reports of CF patients with SARS-CoV-2 from 8 countries. Later, they discuss patients from 4 European countries. Are these countries part of the original 8 countries? Or are they different data. Please correct the text to clarify whether these are common or separate data.

Thank you for this suggestion. This information is now incluted in the revision manuscript (line 266)

The abbreviation CFTR needs to be explained. It should also be explained what the mechanism of CFTR is in SARS-CoV-2 infection/replication. The authors reported that the absence and changes of CFTR affect SARS-CoV-2 replication/infection, but they did not describe the mechanism for this

Thank you for this suggestion. This information is now incluted in the revision manuscript (line 208 and lines 317-19))

Figure 2 missing.

We are sorry for this oversight. Figure 2 is now included in the revised manuscript.

Reviewer 2 Report

In the manuscript titled ‘SARS-CoV-2 infection in patients with cystic fibrosis: what we know so far’, the authors attempted to summarize the status of SARS-CoV-2 infection status in patients suffering from a genetic respiratory disorder cystic fibrosis. The manuscript provides an introduction to the COVID-19 disease epidemiology and its features along with the method of literature review with further subsections stating the descriptions of virulence and molecular features of SARS-CoV-2. The manuscript also illustrates the viral infections in the case of cystic fibrosis and at the end gives an overview of COVID-19 in patients with cystic fibrosis. The article contains an overview of the infection and features of SARS-CoV-2, however, the relevant data along with conclusions of the study especially the claims in the abstract need extensive improvement. The review is supposed to be describing the overview and current status of SARS-CoV-2 in patients with cystic fibrosis. However, the lead-up to the relevant section is extensively described with the main part only confined to one section, which could have been expanded further. Therefore, the study requires significant changes and should be revised for publication in the journal Life.

Critique:

Abstract

The following statement in the abstract is misleading and should be revised to emphasize that protective strategies against their disorder can help avoid COVID-19.

‘such as CF patients, who have strengthened protective strategies to avoid contracting the infection'

The following statement in the abstract is misleading and should be revised

‘to date it is unclear whether cystic fibrosis (CF) patients are at greater risk of COVID-19 or its adverse consequences’

as the following claim of the study from the ‘conclusion’ section contradicts the aforementioned statement.

‘Overall, CF patients with poorer lung function and post-transplant status are at higher risk for worse clinical outcomes in coronavirus disease.’

Introduction

In the introduction, the epidemiological features (initial) of COVID-19 are outdated or should be presented in phrases like ‘one year after the pandemic/six months into the pandemic’ for clear understanding.

There are many grammatical errors throughout the manuscript and need rectification by a thorough proofreading of the article.

The introduction of cystic fibrosis can be initiated separately in a new paragraph.

Methods

The number of studies included in the review has not been mentioned. Since there are only a handful of the retrospective studies mentioned in the last section ‘SARS-CoV-2 infection in cystic fibrosis’, therefore these should be mentioned as it is the only data that is relevant to the title of the manuscript.

SARS-CoV-2 infection in cystic fibrosis

The section needs to be divided further into subsections such as incidence/mortality features of COVID-19 in CF patients.

The data obtained from different retrospective studies can be tabulated with studied parameters aligned with the number of cases to understand the true status of the information.

The factors that can impact the lower incidences such as therapies for CF or increased care for CF patients, need to be separated.

figure 2 is completely missing from the manuscript (only Legend is presented). This oversight itself is enough for rejection, however, a thorough structural change is required for the manuscript.

Conclusion

Each claim of the study needs to be in alignment with the findings and recommendations of the retrospective studies or major highlights of the literature review, therefore it is suggested to include the claims/summary of recommendations in the abstract of the manuscript.

Author Response

We would like to thank the reviewers since their comments have greatly helped us to improve the manuscript which was carefully revised according to their suggestions. 

Abstract

  • The following statement in the abstract is misleading and should be revised to emphasize that protective strategies against their disorder can help avoid COVID-19.

‘such as CF patients, who have strengthened protective strategies to avoid contracting the infection'

Thank you for this suggestion. This statement has been corrected

  • The following statement in the abstract is misleading and should be revised

‘to date it is unclear whether cystic fibrosis (CF) patients are at greater risk of COVID-19 or its adverse consequences’

Thank you for this suggestion. This statement has been corrected (lines 34-35)

as the following claim of the study from the ‘conclusion’ section contradicts the aforementioned statement.

‘Overall, CF patients with poorer lung function and post-transplant status are at higher risk for worse clinical outcomes in coronavirus disease.’

Please see the above response.

Introduction

  • In the introduction, the epidemiological features (initial) of COVID-19 are outdated or should be presented in phrases like ‘one year after the pandemic/six months into the pandemic’ for clear understanding.

Thank you for this suggestion. This information is now incluted in the revision manuscript (lines 54-55)

  • There are many grammatical errors throughout the manuscript and need rectification by a thorough proofreading of the article.

We have carefully proofread our manuscript and corrected grammatical errors.

  • The introduction of cystic fibrosis can be initiated separately in a new paragraph.

Thank you for this suggestion. Cystic fibrosis lung disease has been placed in a new paragraph

Methods

  • The number of studies included in the review has not been mentioned. Since there are only a handful of the retrospective studies mentioned in the last section ‘SARS-CoV-2 infection in cystic fibrosis’, therefore these should be mentioned as it is the only data that is relevant to the title of the manuscript.

SARS-CoV-2 infection in cystic fibrosis

Thank you for this suggestion. This information is now incluted in the revision manuscript (line 114)

  • The section needs to be divided further into subsections such as incidence/mortality features of COVID-19 in CF patients.

We thank you for this suggestion, but as the referee pointed out earlier there are only three retrospective papers, so this kind of analysis, which is already present in the three reported papers, seems redundant, and if the referee agrees we believe that critically reporting the results of these studies is sufficient to understand the true state of information.

  • The data obtained from different retrospective studies can be tabulated with studied parameters aligned with the number of cases to understand the true status of the information.

These data have already been critically analyzed in the papers reported in this Review (see previous response).

  • The factors that can impact the lower incidences such as therapies for CF or increased care for CF patients, need to be separated.

These data have already been critically analyzed in the papers reported in this Review (see previous response).

  • figure 2 is completely missing from the manuscript (only Legend is presented). This oversight itself is enough for rejection, however, a thorough structural change is required for the manuscript.

We apologize for the error, which we realized soon after submitting the work. In agreement with the editor, we sent the missing figure via e-mail. Figure 2 is now included in the revised manuscript.

Conclusion

  • Each claim of the study needs to be in alignment with the findings and recommendations of the retrospective studies or major highlights of the literature review, therefore it is suggested to include the claims/summary of recommendations in the abstract of the manuscript.

Thank you for this suggestion. This information is now incluted in the revision manuscript (lines 39-41)

Reviewer 3 Report

1. It needs to include impact of COVID-19 in health professionals as https://doi.org/10.1007/s11469-020-00418-6 and https://doi.org/10.1136/bmj.m1211 

2. The figure 2 is not include in the manuscript.

3. This phrase "Respiratory infections are the most common and most frequent diseases especially in children 25 and the elderly, characterized by a clear seasonality and with an incidence that usually tends 26 to decrease with increasing age" need a cite.

4. What is the reason to choose August 2020 - September 2022?

5. What is the reason to choose Pubmed? Why not Pubmed and Scopus? or Pubmed and Web of Science?

Author Response

We would like to thank the reviewers since their comments have greatly helped us to improve the manuscript which was carefully revised according to their suggestions.

  1. It needs to include impact of COVID-19 in health professionals as https://doi.org/10.1007/s11469-020-00418-6

and https://doi.org/10.1136/bmj.m1211.

Thank you for this suggestion. This information is now incluted in the revision manuscript (lines 158-160)

  1. The figure 2 is not include in the manuscript.

We apologize for the error, which we realized soon after submitting the work. In agreement with the editor, we sent the missing figure via e-mail. Figure 2 is now included in the revised manuscript.

  1. This phrase "Respiratory infections are the most common and most frequent diseases especially in children and the elderly, characterized by a clear seasonality and with an incidence that usually tends to decrease with increasing age" need a cite.

We agree with the referee, but we did not add the reference because the sentence is in the abstract section.

  1. What is the reason to choose August 2020 - September 2022?

We chose this period to include the first articles published with data on the first wave of the pandemic up to the last articles before the submission of this review.

5. What is the reason to choose Pubmed? Why not Pubmed and Scopus? or Pubmed and Web of Science?

We also searched for articles on Scopus and Web of Science, but found no articles other than those published on Pubmed in these two databases.

Round 2

Reviewer 2 Report

The manuscript has been improved.